# Characterization of the Population of Ovarian Preantral Follicles in Juvenile Six-Banded Armadillos Infected or Not by *Mycobacterium leprae*

**DOI:** 10.3390/vetsci12010037

**Published:** 2025-01-09

**Authors:** Gabriela L. Lima, Andreza V. Brasil, Andreia M. Silva, João Marcelo A. de P. Antunes, Pierre Comizzoli, Alexandre R. Silva

**Affiliations:** 1Federal Institute of Education, Science and Technology of the Ceará State—IFCE, Crato 63100, Brazil; gabriela.lima@ifce.edu.br; 2Laboratory on Animal Germplasm Conservation, Federal University of Semiarid Region—UFERSA, Mossoró 59600, Brazil; andreza.brasil@alunos.ufersa.edu.br (A.V.B.); andreiam.silva@alunos.ufersa.edu.br (A.M.S.); 3Veterinary Hospital, Federal University of Semiarid Region—UFERSA, Mossoró 59600, Brazil; joao.antunes@ufersa.edu.br; 4Smithsonian’s National Zoo and Conservation Biology Institute, Washington, DC 20008, USA; comizzolip@si.edu

**Keywords:** Xenartrhra, Edentata, leprosy, ovary, wildlife

## Abstract

Despite the ecological and public health importance of armadillos, the species is still poorly studied, especially regarding their reproductive aspects. In this work, we present the qualitative characteristics of preantral ovarian follicular population in juvenile six-banded armadillos infected or not by *Mycobacterium leprae*.

## 1. Introduction

The six-banded armadillo (*Euphractus sexcinctus* Linnaeus, 1758) is a wild mammalian species belonging to the superorder Xenarthra that has considerable biological, socioeconomic and public health importance [1,2]. Armadillos play a role in public health since they function as reservoirs of *Mycobacterium leprae*, the infectious agent of leprosy. In Brazil, the country with the highest incidence of leprosis in the Americas [3], six-banded armadillos have been identified as natural carriers of *M. leprae* [4]. Despite this, the species plays a significant role in the balance of the ecosystems it inhabits, as it is responsible for controlling insects through its diet, as well as for soil aeration processes [1].

Although the species is classified as Least Concern [5], it is now facing threats and population decline in Brazil, due to intense hunting pressure, habitat loss and road kills [6,7]. However, conservation strategies are hampered by the lack of basic physiological data also for the *E. sexcintus.* Therefore, the development of conservation strategies has become a priority, including the study of its physiology [8] and conservation breeding strategies [9]. Importantly, approaches developed for the six-banded armadillo could be adapted to other related species, such as the three-banded armadillo (*Tolypeutes tricinctus*) [10] and the giant armadillo (*Priodontes maximus*) [11].

Many aspects of the reproductive physiology of the species still need to be elucidated, which has encouraged specific studies over the years [12]. Work aimed at collecting and conserving male gametes from six-banded armadillos has already been reported [13,14,15]. However, knowledge about female reproduction is limited to monitoring the estrous cycle, which has an average duration of 23.5 days [16]. The female reaches sexual maturity at 274 days of age and is capable of giving birth to two to three young after a gestation period of approximately 68 days [12]. Thus, knowing more about quantitative and qualitative aspects of ovarian anatomy and the follicular population is essential for future development of in vitro oocyte growth and maturation, gonadal tissue cryopreservation and transplantation protocols.

The aim of the study was to characterize and estimate the ovarian preantral follicle population in juvenile six-banded armadillos. In addition, we had the opportunity to describe the ovarian features in individuals infected or not infected with *M. leprae*.

## 2. Materials and Methods

### 2.1. Source of Animals

The animals used in the present study were part of an epidemiological investigation of the presence of *M. leprae* in six-banded armadillos in the state of Rio Grande do Norte, Brazil. The complete data from this investigation are published in the work of da Silva Ferreira et al. [3]. Five female six-banded armadillos (*Euphractus sexcinctus*) were obtained alive by wildlife veterinarians from separate locations near the city of Mossoró, Rio Grande do Norte state, Brazil, in May 2016. The animals were then characterized as young based on their body weight and based on the histological characteristics of their ovary relative to the emergence of secondary follicles, as previously described for another species of armadillo, *Dasypus novemcinctus* [17]. Here, we remember that, to date, there are no indications that the species presents any type of reproductive seasonality, in intertropical regions such as the one where the present work was conducted. At the UFERSA veterinary hospital, the animals were weighed and anesthetized with an intramuscular administration of tiletamine-zolazepam (Zoletil^®^ 50, Virbac, Brazil), at a dose of 4 mg/kg to allow visual inspection of the entire body for the presence of lesions indicative of *M. leprae* infection as recommended by Sharma et al. [18]. The animals’ blood was collected by cardiac puncture, using a 10 mL syringe with a 22 G needle. Then, euthanasia was performed by the administration of potassium chloride via the femoral vein at a dose of 2.56 mEq/Kg (1 mL/kg of a 19.1% solution).

### 2.2. Detection of M. leprae Infection

Briefly, serum was obtained from the animals’ blood samples and aliquoted in volumes of 100 μL into Eppendorf centrifuge tubes (1.5 mL volume), which were frozen in dry ice for transport to the Laboratory of Cellular Microbiology (LCM), Instituto Oswaldo Cruz, FIOCRUZ, Rio de Janeiro. The tubes were stored in a freezer (−20 °C) until analysis. Serum was collected and examined using two “in-house” enzyme-linked immunosorbent assays (ELISAs) and via two commercially available (ML flow and NDO-LID^®^) immunochromatographic lateral flow (LF) tests, for detection of the PGL-I and/or LID-1 antigens of the bacterium [3]. Immunoenzymatic tests revealed that of the five animals, two were infected with *M. leprae*. Thus, the results of the study will be presented considering the uninfected animals, the infected animals, and the total values for the species.

### 2.3. Measurements of the Ovaries

After euthanasia, each individual’s pair of ovaries was collected and washed in 70% alcohol for 10 s, followed by two rinses in saline solution for 10 s each to eliminate blood and other contaminants. The ovaries were then measured for width, length, and thickness using a caliper and weighed on a digital scale. The gonadosomatic index (GSI) was determined based on the equation GSI = ovarian weight/animal weight × 100 [19].

### 2.4. Histological Processing

After processing, each pair of ovaries was fixed in Carnoy’s solution for 12 h, then dehydrated in a series of increasing ethanol concentrations, cleared in xylene, and embedded in histological paraffin. The resulting paraffin blocks were sectioned in a 5 μm series. Every 120th section was mounted on slides and stained with hematoxylin-eosin [20]. A mean of 180 and 188 histological sections were obtained from the right and left ovaries, respectively, and two sections per ovary were evaluated for the ovarian preantral follicle estimation.

### 2.5. Ovarian Preantral Follicles Morphometrics

For morphometry analysis, approximately 30 follicles were evaluated for each category (primordial, primary, and secondary) by light microscopy, under 400× or 1000× magnification (Zeiss, Muenchen, Germany), when microphotographs were taken. The mean of minimum and maximum (μm) diameters of the follicles, oocytes and nucleus from primordial, primary and secondary follicles were taken with the Image J free software (Version 1.54) [21].

### 2.6. Estimation of Ovarian Preantral Follicle Population

Follicles were identified and classified as primordial when they had an oocyte with a visible nucleus and were surrounded by a single layer of granulosa cells in a squamous shape; primary when they contained an oocyte surrounded by a layer of cubic-shaped cells; and secondary when the oocyte present was surrounded by two or more layers of cubic-shaped cells, without the presence of an antral cavity [22].

Preantral follicles (PFs) were counted according to their category and the numbers obtained were applied to the formula to estimate the ovarian preantral follicle (PF) population according to Gougeon and Chainy [20]:PF Population = Nº of follicles × Nº of sections obtained × Thickness of sectionsNº of sections observed × Mean of oocyte nuclei diameter

### 2.7. Morphological Analysis of Ovarian Preantral Follicles

The preantral follicles (PFs) were identified and classified under light microscopy at 1000× magnification based on their structural integrity. Follicular morphology was assessed by examining the integrity of the oocyte, granulosa cells, and basement membrane. PFs were categorized as either morphologically normal or degenerated. Normal follicles contained an oocyte with a regular shape, uniform cytoplasm, and organized layers of granulosa cells. Degenerated follicles showed an oocyte with a pyknotic nucleus and/or shrunken ooplasm, in some cases, granulosa cell layers were disorganized, detached from the basement membrane, or included enlarged cells. To prevent duplicate evaluations, only sections with a visible oocyte nucleus were analyzed to count preantral follicles [22].

## 3. Results

### 3.1. Overall Anatomy of the Ovaries

The juvenile six-banded armadillo ovaries are round structures with smooth surfaces (Figure 1).

Values for body weight, ovary weight, ovary dimensions (width × length × thickness) and gonadosomatic index in six-banded armadillos infected (n = 2) or not (n = 3) by the *Mycobacterium leprae* are presented in Table 1.

### 3.2. Histological Features of the Ovarian Tissues and Preantral Follicles

Microscopical analysis showed that ovaries contained a medulla and a cortical region with follicles at various stages of development (Figure 2). The primordial and primary follicles were located close to the germinative epithelium (superficial margins of the ovarian cortex), while the secondary ones were located in the inner part.

Primordial and primary follicles were characterized by an oocyte surrounded by a single layer of flattened or cuboidal granulosa cells, respectively (Figure 3a,b). Secondary follicles had two or more concentric layers of cuboidal granulosa cells around the oocyte. Externally of the granulosa cells, the internal theca was formed by elongated and fusiform cells, appearing as concentric layers. An evident zona pellucida was observed between granulosa cell layers and oocyte (Figure 3c). Oocytes presented heterogenic cytoplasm with some vacuoles. The nuclear membrane was visible. Scattered regions of chromatin and heterochromatin were located near the nucleus membrane. Multi-oocytes follicles (Figure 3d) were observed in all animals, near the germinative epithelium. Overall, a total of 4 to 150 multi-oocyte follicles per ovarian pair were observed (Table 2).

### 3.3. Ovarian Preantral Follicles Morphometrics and Estimation

Data related to preantral follicle morphometry are presented in Table 3.

### 3.4. Estimation of the Ovarian Preantral Follicle Population

Considering the five individuals, it could be inferred that the follicular population of the species would be 22,996.2 ± 7541.6 follicles per ovarian pair, as demonstrated in Table 4. When evaluating the data per animal in detail, however, we verified that individual A5 has a follicular population more than twice that of the other individuals. Therefore, when recalculating the follicular population based only on the other four individuals, we believe that the value of 15,567.2 preantral follicles per ovarian pair is more realistic to characterize the estimate of the follicular population of the species. Moreover, for four females, most of the follicular population was formed by primordial follicles, except in female A1, which presented a larger population of primary follicles.

### 3.5. Morphology of the Ovarian Preantral Follicles

Regardless of *M. leprae* infection, a high rate of ovarian follicle degeneration was observed in *E. sexcinctus* females, since only 41.4 ± 3.2% of follicles were classified as morphologically normal (Table 5).

Morphologically normal PFs showed a spherical oocyte with an eosinophilic nucleus and heterogeneous cytoplasm. Granulosa cells without pycnotic nuclei were well-organized in layers surrounding the oocyte. Degenerated follicles showed a retraced oocyte with or without a pycnotic nucleus, accompanied or not by disorganized granulosa cells (Figure 4).

## 4. Discussion

The study provided important data on the characterization and estimation of the ovarian anatomy and preantral follicle populations in juvenile six-banded armadillos. Campos et al. [16] previously described the six-banded armadillo ovary appearance by ultrasound. The female gonads were well-defined structures, rounded and slightly hypoechoic in relation to the adjacent tissue. The ovary length measurement described by the authors was the same in the present study (0.90 cm); however, the width was larger (0.4 cm). It should be noted that during the process of forming the ultrasound image, there is the occurrence of the formation of image artifacts, which refer to the projection of images that do not match exactly with the true image of the location examined, mis-estimating the dimension of the ovary [16].

The ovarian measurements of the six-banded armadillo are similar to those of the lesser hairy (*Chaetophractus vellerosus*—0.5 cm) and the pichi (*Zaedyus pichiy*—0.4 cm) armadillos, larger than those of the pink fairy armadillo (*Clamyphorus truncatus*—0.2 to 0.3 cm), but smaller than those of the Azara’s domed armadillo (*Tolypeutes matacus*—0.8 to 1.0 cm), the large hairy armadillo (*Chaetophractus villosus*—0.6 to 1.0 cm) and the southern lesser long-nosed armadillo (*Dasypus hybridus*—0.6 cm to 0.9 cm) [23]. Studies have shown variations among the female reproductive organs of Dasypodidae species, mainly those related to the ovaries, uterus, and the lower portion of the reproductive tract [24,25,26]. These differences may be linked to the diversity of sperm types observed in Dasypodidae. In this family, the structure of the female genital tracts may create distinct functional and structural barriers for sperm, potentially leading to changes in sperm shape and size, suggesting a coevolutionary process between female reproductive traits and male gamete traits [23].

A large individual variation was verified for GSI (4.81–10.0) in juvenile *E. sexcinctus*. For other armadillos, data regarding GSI are not described, making it impossible to compare the results obtained with those of other species. In fact, GSI is not commonly described for mammalian females [19]. On the other hand, it is well described for aquatic species, such as fish and mollusks, being fundamental to understanding its reproductive status, oocyte size and ovarian maturity [27].

Microscopically, the analysis showed that the ovaries of six-banded armadillos were divided into a medulla and a cortical region, where ovarian follicles at various stages of development were found. Ovarian follicles are distributed in the ovarian cortical zone and are formed by an oocyte surrounded by a layer of flattened granulosa cells or one or more cuboidal cells, depending on the stage of follicle development. These characteristics are similar to those described for free-living six-banded armadillos from the central-western or southern regions of Brazil [8].

The follicular classification adopted in this study was in accordance with Silva et al. [22], classifying preantral follicles into three stages: primordial, primary, and secondary. This classification is different from that adopted by Rezende et al. [8] for the armadillo, in which preantral follicles were categorized as primordial, primary one-layer, primary two-layer, and secondary (with the initial formation of the antrum cavity). Despite the divergence in follicular classification, similarities were observed regarding the characteristics of the follicles described. In both studies, vacuoles were observed in the cytoplasm of oocytes, which presented evident nuclear membrane, with dispersed regions of chromatin and heterochromatin. Furthermore, the characterization of ovarian follicles has been previously performed in other armadillo species. The same ovarian follicle classification and similar characteristics were also described for the pichi, the lesser hairy and the southern lesser long-nosed armadillos [23]. In the hairy armadillo, for example, ovarian follicles were categorized as primordial, intermediate, early primary, late primary, secondary, tertiary, and Graafian or preovulatory [28] or into four stages (I–IV), based on follicle size and granulosa cell layers and shape [29].

As described here for the six-banded armadillo, the occurrence of multi-oocyte follicles was also observed for other species of the families Euphractinae (*Ch. villosus*, *Ch. vellerosus*, *Z. pichiy* and *C. truncatus*) and Tolypeutinae (*T. matacus*) [23,25,30]. In nine-banded armadillos, even when large follicles (>978 μm) were present, indicating proximity to ovulation, nests of primordial follicles were still found in the ovaries [17]. These multi-oocyte follicles were observed in all six-banded armadillos used in the present study, while in other species, such as *Z. pichiy*, it was not observed in all animals studied, inferring an interspecies variation [30]. The high variation of the multi-oocyte number observed among animals, both infected or not, may be attributed to the individual variation, as poly-ovular development seems not to be only a natural polymorphism [30], causing alterations of nest breakdown, but also is related to environmental endocrine disruptors and phytoestrogens [31]. All those factors may have contributed to the large individual difference observed among animals.

The reason for the occurrence of this phenomenon has not yet been completely elucidated. Theories have been suggested to explain its existence, such as mitotic division with a high rate of cell differentiation during follicle growth, from which several oocytes arise from primordial follicles [32], a failure of cell division in the initial stages of follicle development fusion of adjacent individual follicles [31].

In the large hairy armadillo, after light microscopy, the presence of multi-oocyte follicle-like structures was observed; however, ultrastructural and immunohistochemistry analysis showed the presence of ovarian germ cell (GC) cysts, interconnected by intercellular bridges and surrounded by a single layer of flat follicle cells [33]. In mammals, the GC cyst rupture process usually concludes around birth or shortly thereafter, with the development of primordial follicles containing single oocytes, which form the only gamete reservoir available throughout the female’s reproductive life [34]. The existence of GC cysts in adult animals is an evolutionary-conserved developmental event that enables a deeper investigation of the events surrounding folliculogenesis [33]. Further investigation is required to affirm if the structures classified as multi-oocyte follicles in six-banded armadillos are muti-oocytes or GC cysts involved in the formation of new primordial follicles after animal birth.

In the large hairy armadillo, using transmission electron microscopy, a distinct structure was described in the cytoplasm of oocytes in early growth, the multilamellar body—MLBs. The authors described this structure as an organelle with a transitory function, during the initial stages of follicular development, from follicle growth to the beginning of zona pellucida formation [29]. The MLBs are composed of lamellar units different from filaments, and these lamellae may coalesce and form large lamellar patches. A vital role of this organelle to the zona pellucida development is suggested. Some of the vacuoles observed in the six-banded armadillo’s oocyte cytoplasm may be related to the presence of the MLBs or to lipid droplets, as is observed in other mammals, such as peccaries [35]. However, further studies using transmission electron microscopy are required to confirm the hypothesis.

Regarding morphometry, the values obtained in the present work were closer to those described for a large hairy armadillo. Similarly, the oocytes increased their diameter according to the development of the follicle, while a smaller nuclear increase was observed [29]. Moreover, a positive and linear correlation between the diameter of the follicle and the oocyte was verified for large hairy armadillos [28]. Previous studies showed differences among the number and size of armadillos’ follicles throughout the year, indicating seasonal changes [17,33]. This can be related to the differences observed in this work and other studies. However, the existence of seasonal reproductive changes in the six-banded armadillo still needs to be investigated.

The follicular population observed here for six-banded armadillos was considerably larger than that observed in the greater hairy armadillo [33], in which the number of primordial follicles ranged from 1107.9 ± 362.4 (summer) to 2990.3 ± 306.3 (autumn), with the same number observed for primary follicles (257.5 ± 40.1 to 582.7 ± 33). It is worth highlighting that, in the present study, a formula was used to estimate the follicular population that has been previously validated and extensively reviewed for other domestic [36,37] and wild [35,38] species, as well as for humans [20]. As in the present study, in sloths (*Bradypus variegatus*), the majority of follicles found were characterized as primordial, forming the ovarian reserve pool [39].

In all-female six-banded armadillos, a high rate of ovarian follicular degeneration was observed, especially when compared with values found for other mammalian species [35,36,37]. Follicular development is known to depend on the rates of proliferation and atresia, which are regulated by many endocrine, paracrine, and autocrine factors [40]. Although atresia leads to the loss of numerous ovarian follicles, it plays a vital role in maintaining ovarian homeostasis in mammals, ensuring regular reproductive cyclicity [41]. In six-banded armadillos, however, further investigation is needed to elucidate whether such a high rate of degenerated follicles is physiological or related to external factors.

Regarding infection by *M. leprae*, no major changes were observed in the parameters evaluated in the ovary of six-banded armadillos. This information contradicts what was previously described for the nine-banded armadillo, in which the occurrence of considerable lepromatous infiltration in the ovaries of experimentally infected individuals was observed [42]. We, however, recognize the limitations of our findings, given the limited number of animals used. In any case, since this is a wild animal, the importance of the data should be considered, especially given the difficulties in obtaining access to such animals. This difficulty is also evident in the study that demonstrated for the first time the effect of *M. leprae* infection on the pregnancy of nine-banded armadillos, whose report was based on the description of three cases [43].

## 5. Conclusions

We demonstrated that the juvenile six-banded armadillo presents particularities regarding the characteristics and estimation of its ovarian preantral follicular population. Furthermore, we report that *M. leprae* infection apparently does not cause major changes in histological ovarian features in this species. With these data, we highlight that folliculogenesis in Xenarthra remains largely unexplored and studying it can provide valuable data on reproductive physiology, improving captive management, epidemiological studies and assisted reproductive technologies.

## Figures and Tables

**Figure 1 vetsci-12-00037-f001:**
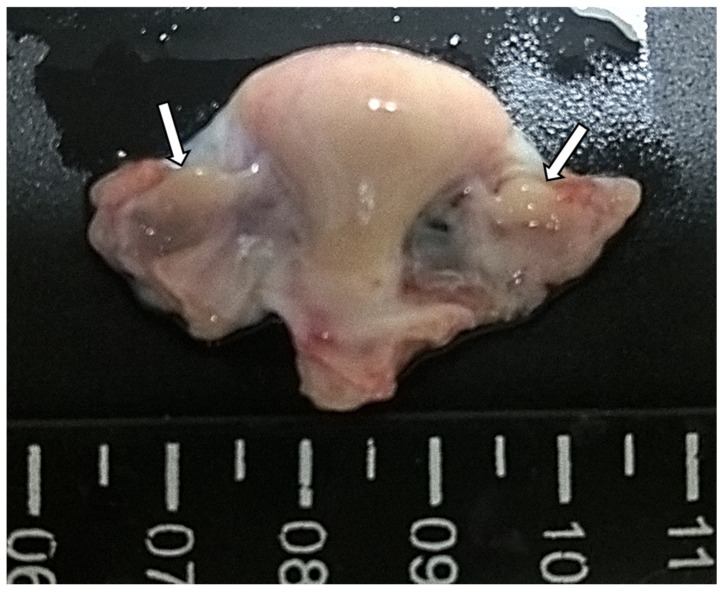
Macroscopic aspects of *Euphractus sexcintus* reproductive system (uterine body in the middle and ovaries on each side). Ovaries are round structures with smooth surfaces (white full arrows).

**Figure 2 vetsci-12-00037-f002:**
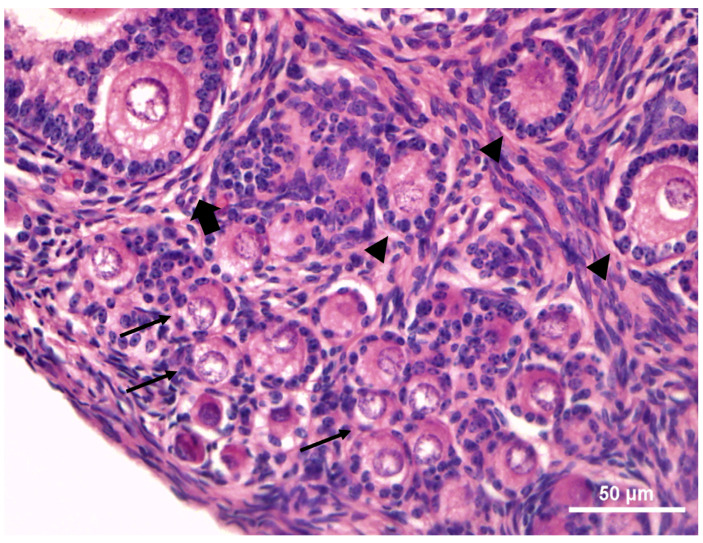
Histology of *Euphractus sexcintus* ovary with follicles at various stages of development. Primordial (arrows), primary (arrowhead) and secondary (full arrow) follicles. 400×.

**Figure 3 vetsci-12-00037-f003:**
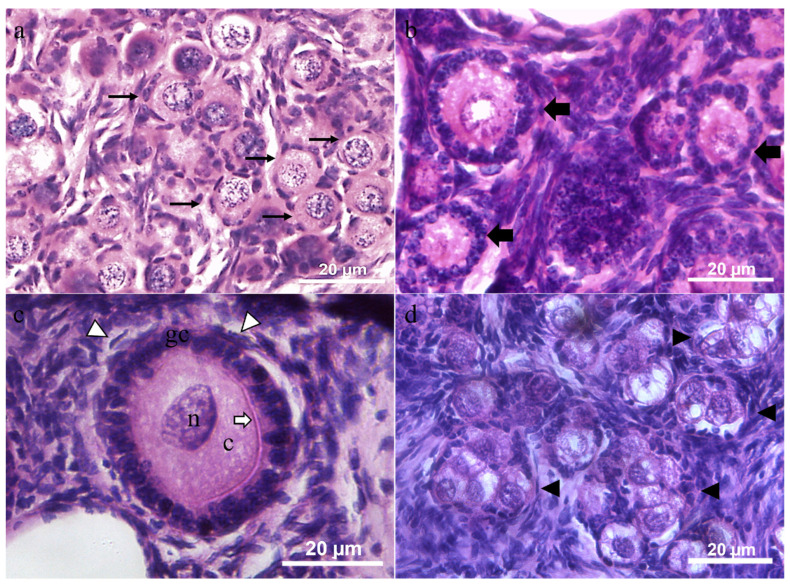
Histological features in the juvenile six-banded armadillos (*Euphractus sexcintus)* preantral follicles. (**a**)—Primordial follicles (arrows) surrounded by a single layer of flattened granulosa cells. (**b**)—Primary follicles (full arrows) surrounded by a single layer of cuboidal granulosa cells, presenting some vacuoles at cytoplasm. (**c**)—Small secondary follicle showing two or more layers of cuboidal granulosa cells (n—nucleus, c—cytoplasm, gc—granulosa cells) and the presence of zona pellucida (white arrow). The theca cells were formed by elongated and fusiform cells (white arrowheads). (**d**)—multi-oocyte follicles (arrowheads). 400×.

**Figure 4 vetsci-12-00037-f004:**
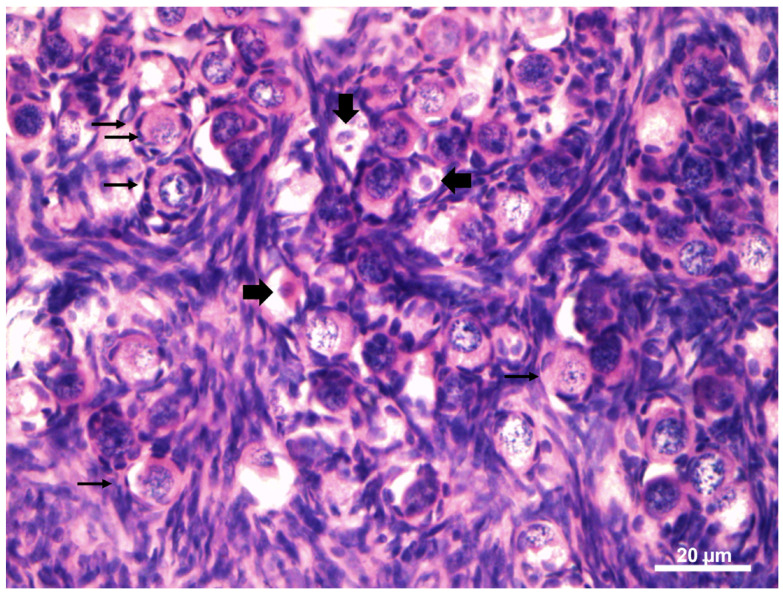
Morphological normal follicles (arrows) exhibiting well-organized granulosa cells without pycnotic nuclei surrounding the oocyte. Degenerated follicles (full arrows) showing a retraced oocyte with a pycnotic nucleus, accompanied or not by disorganized granulosa cells. 400×.

**Table 1 vetsci-12-00037-t001:** Body weight, ovary weight, ovary dimensions and gonadosomatic index in juvenile six-banded armadillos infected (n = 2) or not (n = 3) by the *Mycobacterium leprae*.

Animals	Health Condition	Body Weight (Kg)	Right Ovary		Left Ovary		Gonadosomatic Index (%)
			Weight (Kg)	Dimensions (mm) (Width × Length × Thickness)	Weight (Kg)	Dimensions (mm) (Width × Length × Thickness)	
A1	Infected	1.83	0.06	0.4 × 1 × 0.3	0.05	0.4 × 0.8 × 0.3	6.01
A2	Infected	1.93	0.07	0.4 × 1 × 0.2	0.06	0.4 × 1 × 0.2	6.74
A3	Uninfected	2.08	0.05	0.5 × 0.8 × 0.2	0.05	0.4 × 1 × 0.3	4.81
A4	Uninfected	1.60	0.06	0.4 × 0.8 × 0.3	0.10	0.5 × 1 × 0.3	10.0
A5	Uninfected	1.70	0.06	0.4 × 0.9 × 0.3	0.06	0.4 × 0.9 × 0.3	7.06
Means ± SEM		1.8 ± 0.1	0.1 ± 0.0	0.4 × 0.9 × 0.3	0.1 ± 0.0	0.4 × 0.9 × 0.3	6.9 ± 0.8

**Table 2 vetsci-12-00037-t002:** Number of multi-oocyte preantral follicles found in the ovarian pair of juvenile six-banded armadillos infected (n = 2) or not (n = 3) by the *Mycobacterium leprae*.

Animals	Health Condition	Right Ovary	Left Ovary	Total per Ovarian Pair
A1	Infected	7	6	13
A2	Infected	99	51	150
A3	Non-infected	28	12	40
A4	Non-infected	2	2	4
A5	Non-infected	70	22	92
Means ± SEM		37.2 ± 18.8	18.6 ± 8.8	59.8 ± 27.3

**Table 3 vetsci-12-00037-t003:** Morphometry (μm) means ± SEM of the oocyte nucleus, oocyte, and follicles (per category) in the ovary of juvenile six-banded armadillos infected (n = 2) or not (n = 3) by the *Mycobacterium leprae*.

Animals	Health Condition	Follicle Category	Nucleus	Oocyte	Follicle	Number of Follicles Evaluated
A1	Infected	Primordial	8.0 ± 0.3	13.4 ± 0.3	17.2 ± 0.3	30
		Primary	9.0 ± 0.3	16.3 ± 0.5	23.4 ± 0.9	30
		Secondary	12.9 ± 0.4	26.9 ± 1.3	42.9 ± 1.8	28
A2	Infected	Primordial	7.2 ± 0.3	12.4 ± 0.2	16.1 ± 0.3	30
		Primary	8.9 ± 0.2	16.3 ± 0.4	23.5 ± 0.4	30
		Secondary	13.8 ± 1.2	32.5 ± 5.3	47.9 ± 7.3	10
A3	Non-infected	Primordial	7.5 ± 0.1	10.5 ± 0.1	13.8 ± 0.2	30
		Primary	7.1 ± 0.2	10.2 ± 0.2	14.3 ± 0.2	30
		Secondary	8.1 ± 0.2	12.4 ± 0.8	18.7 ± 0.7	30
A4	Non-infected	Primordial	6.9 ± 0.2	10.7 ± 0.2	14.4 ± 0.2	30
		Primary	8.7 ± 0.2	17.5 ± 0.5	24.2 ± 0.7	30
		Secondary	10.3 ± 1.3	24.0 ± 4.1	34.8 ± 7.9	10
A5	Non-infected	Primordial	5.6 ± 0.1	8.5 ± 0.2	11.7 ± 0.2	30
		Primary	7.1 ± 0.3	14.1 ± 0.7	21.8 ± 1.1	29
		Secondary	9.3 ± 0.5	22.8 ± 1.4	38.0 ± 2.0	28
Means ± SEM		Primordial	7.0 ± 0.2	11.1 ± 0.2	14.6 ± 0.2	150
		Primary	8.2 ± 0.2	14.9 ± 0.5	21.4 ± 0.7	149
		Secondary	10.9 ± 0.7	23.7 ± 2.6	36.5 ± 3.9	106

**Table 4 vetsci-12-00037-t004:** Estimation of preantral follicle (PF) population per ovary (right and left) and per ovarian pair (Total PF Population) of juvenile six-banded armadillos infected (n = 2) or not (n = 3) by the *Mycobacterium leprae*.

Animals	Health Condition	Follicle Category	PF Population of Right Ovary	PF Population of Left Ovary	Total PF Population	Proportions (%) of PF Category
A1	Infected	Primordial	2929.2	2484.9	5414.1	35.4
		Primary	5371.9	3697.1	9068.9	59.4
		Secondary	315.3	476.9	792.18	5.2
		Total	8616.4	9143.7	17,760.1	100.0
A2	Infected	Primordial	7210.8	3708.7	10,919.4	64.4
		Primary	3222.6	2214.8	5437.4	32.1
		Secondary	287	302.9	589.8	3.5
		Total	10,720.3	6226.3	16,946.7	100.0
A3	Uninfected	Primordial	4559.18	7168.6	11,727.8	69.0
		Primary	2517.8	2562.2	5082.0	29.9
		Secondary	120.5	61.8	182.3	1.1
		Total	7197.4	9792.7	16,990.1	100.0
A4	Uninfected	Primordial	4598.9	2426.6	7025.6	66.5
		Primary	1490.7	1908.0	3398.7	32.2
		Secondary	57.8	89.7	147.4	1.3
		Total	6147.4	4424.3	10,571.8	100.0
A5	Uninfected	Primordial	20,368.9	7628.7	27,997.5	53.1
		Primary	14,886.6	8812.7	23,699.3	44.9
		Secondary	756.0	259.5	1015.6	2.0
		Total	36,011.5	16,700.9	52,712.4	100.0
Means ± SEM			13,738.6 ± 5620.7	9257.6 ± 2100.6	22,996.2 ± 7541.6	-

**Table 5 vetsci-12-00037-t005:** Morphological integrity of preantral follicles in the ovaries of juvenile six-banded armadillos infected (n = 2) or not (n = 3) by the *Mycobacterium leprae*.

			Total			Proportions (%)	
Animals	Health Condition	Follicle Category	Normal	Degenerated	Total Follicle/ Category	Normal	Degenerated
A1	Infected	Primordial	1334.6	4079.5	5414.1	24.7	75.4
		Primary	3120.8	5948.1	9069.0	34.4	65.6
		Secondary	265.5	526.7	792.2	33.5	66.49
A2	Infected	Primordial	4162.7	6756.7	10,919.4	38.1	61.9
		Primary	2309.9	3127.5	5437.4	42.5	57.5
		Secondary	166.0	423.8	589.8	28.2	71.9
A3	Uninfected	Primordial	6190.8	5536.9	11,727.6	52.8	47.2
		Primary	2748.9	2333.1	5082.0	54.1	45.9
		Secondary	94.3	88	182.3	51.7	48.3
A4	Uninfected	Primordial	3999.6	3026.0	7025.6	56.9	43.1
		Primary	1789.3	1609.5	3398.7	52.7	47.4
		Secondary	20.7	126.7	147.4	14.0	86.0
A5	Uninfected	Primordial	11,482.6	16,514.9	27,997.3	41.0	59.0
		Primary	11,293.9	12,405.4	23,699.3	47.7	52.3
		Secondary	490.9	524.7	1015.6	48.3	51.7
Means ± SEM			3298.0 ± 965.7	4201.8 ± 1230.3	7499.9 ± 2169.6	41.4 ± 3.2	58.6 ± 3.2

## Data Availability

The data are available from the authors upon reasonable request.

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
