# Peer review of "Characterization of the Population of Ovarian Preantral Follicles in Juvenile Six-Banded Armadillos Infected or Not by *Mycobacterium leprae"

_vetsci, 2025, doi:10.3390/vetsci12010037_

Round 1

Reviewer 1 Report

Comments and Suggestions for Authors

The article is interesting from a zoological point of view because it presents previously unexplored facts. One of the problems of this work is the small number of animals studied, which reduces the level of validity of the conclusions. I would rather recommend that the article be published with some corrections.

Figure 1 in not focused. If it is possible, provide another photo.

Fig 3a (Primordial follicles surrounded by a single layer of flattened granulosa cells, presenting some vacuoles at cytoplasm). The vacuoles are not clearly visible. Please, provide another photo.

I think that “multioocytes follicles” should be replaced by “multi-oocyte follicles”.

In Figure 4, I cannot see clear difference between pycnotic and normal nuclei. Please, provide another photo.

Author Response

Referee #1

Comments 1: The article is interesting from a zoological point of view because it presents previously unexplored facts. One of the problems of this work is the small number of animals studied, which reduces the level of validity of the conclusions. I would rather recommend that the article be published with some corrections.

Response 1: We thank the reviewer for the valuable comments. In the discussion, we acknowledge the limits of our conclusions due to the number of individuals involved in the study. Meanwhile, we emphasize the importance of the data, given that this is a wild species that is difficult to access. Even a simple scientific report based on a few individuals provide essential information for a species that has not yet been studied. The study provides unprecedented information regarding the reproductive physiology of six-banded armadillos.

Comments 2: Figure 1 in not focused. If it is possible, provide another photo.

Response 2: We tried to provide a higher resolution image of the animals' reproductive tract, but we were unable to do so. These images were taken on the day the animals were euthanatized, unfortunately, using a low-quality camera. We were not originally planning to include this image in the manuscript, but we decided to keep it because it is informative.

Comments 3: Fig 3a (Primordial follicles surrounded by a single layer of flattened granulosa cells, presenting some vacuoles at cytoplasm). The vacuoles are not clearly visible. Please, provide another photo.

Response 3: We replaced the photo for a better one in which vacuoles are clearly observed.

Comments 4: I think that “multioocytes follicles” should be replaced by “multi-oocyte follicles”.

Response 4: We revised this throughout the text, as suggested.

Comments 5: In Figure 4, I cannot see clear difference between pycnotic and normal nuclei. Please, provide another photo.

Response 5: We replaced the photo by a better one.

Reviewer 2 Report

Comments and Suggestions for Authors

Histological Study Report of Tatu's Ovary with and without Mycobacterium leprae Infection

1.Introduction

In the present study, a histological examination of Tatu's ovary with and without Mycobacterium leprae infection was performed with the aim of evaluating the microscopic characteristics of ovarian tissue, particularly the ovarian follicles. Histological examination is essential for understanding cellular and tissue morphology, allowing for a detailed analysis of alterations and potential pathologies present in the organ.

2. Materials and Methods

The ovarian sample was properly collected following the study's standards for histological evaluation. The analysis was conducted with the aid of an optical microscope, and observations were recorded to identify important aspects of ovarian morphology.

3. Results

Upon examining the histological sections, we observed that the photomicrographs were of excellent quality. The morphometric data were well-defined.

4. Discussion and Conclusion

A highlight of this article is the discussion. With these data, we emphasize that folliculogenesis in armadillos remains largely unexplored and needs to be studied, as it may provide valuable information about reproduction, improving captive management and epidemiological studies.

Author Response

Reviewer #2

Comments 1. Introduction - In the present study, a histological examination of Tatu's ovary with and without Mycobacterium leprae infection was performed with the aim of evaluating the microscopic characteristics of ovarian tissue, particularly the ovarian follicles. Histological examination is essential for understanding cellular and tissue morphology, allowing for a detailed analysis of alterations and potential pathologies present in the organ.

Response 1: The authors thank the reviewer for highlighting the importance of our work and the constructive comments.

Comments 2: Materials and Methods – The ovarian sample was properly collected following the study's standards for histological evaluation. The analysis was conducted with the aid of an optical microscope, and observations were recorded to identify important aspects of ovarian morphology.

Response 2: Thank you.

Comments 3: Results – Upon examining the histological sections, we observed that the photomicrographs were of excellent quality. The morphometric data were well-defined.

Response 3 – Thank you.

Comments 4: Discussion and Conclusion – A highlight of this article is the discussion. With these data, we emphasize that folliculogenesis in armadillos remains largely unexplored and needs to be studied, as it may provide valuable information about reproduction, improving captive management and epidemiological studies.

Response 5: In fact, the study of folliculogenesis in this species still needs to be further investigated. However, we believe that we have taken the first step in this regard, providing important morphophysiological data through the use of histological techniques, which are fundamental for understanding the reproductive physiology of this species.

Reviewer 3 Report

Comments and Suggestions for Authors

The significance of this study is limited due to the small number of experimental numbers.

Author Response

Reviewer #3

Comments 1: The significance of this study is limited due to the small number of experimental numbers.

Response 1: In the discussion, we acknowledge the limits of our conclusions due to the number of individuals used. In the meantime, we emphasize the importance of the data, given that this is a wild species that is difficult to access. We would also like to emphasize that both the Ministry of the Environment and the Ethics Committees strictly limit access to the number of wild animals that can be used in scientific studies. Therefore, we took advantage of animals that were used for an epidemiological study on leprosy. When it comes to wild animals, important information about the biology or physiology of the species often comes only from reports, often from a single or few animals, especially for morphological descriptions, as shown in the following articles:

  1. Lopes GP, Santos RR, Almeida DV, Brito AB, Queiroz HL, Domingues SFS. Population estimate and morphometry of ovarian preantral follicles from three recently recognized squirrel monkey species: a comparative study. Zygote. 2017 Jun;25(3):279-287. doi: 10.1017/S0967199417000107. (They used 01 Saimiri microdon, 02 Saimiri cassiquiarensis, and only one ovary from a Saimiri vansolini)
  2. Radcliffe RW, Eyres AI, Patton ML, Czekala NM, Emslie RH. Ultrasonographic characterization of ovarian events and fetal gestational parameters in two southern black rhinoceros (Diceros bicornis minor) and correlation to fecal progesterone. 2001 Mar 15;55(5):1033-49. doi: 10.1016/s0093-691x(01)00464-2. (They used 02 individuals)
  • Klohonatz K, Durrant B, Sirard MA, Ruggeri E. Granulosa cells provide transcriptomic information on ovarian follicle dynamics in southern white rhinoceros. Sci Rep. 2024 Aug 20;14(1):19321. doi: 10.1038/s41598-024-70235-7. (They used 03 individuals)
  1. Rodrigues FR, Da Silva VM, Barcellos JF, Lazzarini SM. Reproductive anatomy of the female Amazonian manatee Trichechus inunguis Natterer, 1883 (Mammalia: Sirenia). Anat Rec (Hoboken). 2008 May;291(5):557-64. doi: 10.1002/ar.20688. (They used 03 individuals)
  2. Kinoshita K, Nakamura T, Kimura K, Shimizu M, Kuze N, Ozaki Y. Gynaecological diagnosis by ultrasound and the measurement of urinary sex steroid hormones in female orangutans. Vet Med Sci. 2020 Aug;6(3):612-616. doi: 10.1002/vms3.237. E (They used 03 individuals)
  3. Scalercio SR, Brito AB, Domingues SF, Santos RR, Amorim CA. Immunolocalization of growth, inhibitory, and proliferative factors involved in initial ovarian folliculogenesis from adult common squirrel monkey (Saimiri collinsi). Reprod Sci. 2015 Jan;22(1):68-74. doi: 10.1177/1933719114532842. (They used 04 individuals)
  • Sattler R, Bishop A, Woodie K, Polasek L. Characterizing estrus by trans-abdominal ultrasounds, fecal estrone-3-glucuronide, and vaginal cytology in the Steller sea lion (Eumetopias jubatus). 2018 Oct 15;120:25-32. doi: 10.1016/j.theriogenology.2018.07.020. (They used 03 individuals)

In time, we note that, even in domestic species, work related to the characterization of ovarian features and follicular population estimation usually uses few individuals, as stated in the works below:

  1. Carrijo OA Jr, Marinho AP, Campos AA, Amorim CA, Báo SN, Lucci CM. Morphometry, estimation and ultrastructure of ovarian preantral follicle population in queens. Cells Tissues Organs. 2010;191(2):152-60. doi: 10.1159/000225935. (They used 05 queens)
  2. Mondadori RG, Santin TR, Fidelis AA, Porfírio EP, Báo SN. Buffalo (Bubalus bubalis) pre-antral follicle population and ultrastructural characterization of antral follicle oocyte. Reprod Domest Anim. 2010 Feb;45(1):33-7. doi: 10.1111/j.1439-0531.2008.01199.x. (They used 05 female buffalos)
  • Warren L, Murawski M, Wilk K, Zieba DA, Bartlewski PM. Suitability of antral follicle counts and computer-assisted analysis of ultrasonographic and magnetic resonance images for estimating follicular reserve in porcine, ovine and bovine ovaries ex situ. Exp Biol Med (Maywood). 2015 May;240(5):576-84. doi: 10.1177/1535370214560971. (They used 05 porcine, 05 ovine and 05 bovine females)
  1. Lucci CM, Amorim CA, Rodrigues AP, Figueiredo JR, Báo SN, Silva JR, Gonçalves PB. Study of preantral follicle population in situ and after mechanical isolation from caprine ovaries at different reproductive stages. Anim Reprod Sci. 1999 Aug 16;56(3-4):223-36. doi: 10.1016/s0378-4320(99)00045-7. (They used 06 does)
  2. Songsasen N, Fickes A, Pukazhenthi BS, Wildt DE. Follicular morphology, oocyte diameter and localisation of fibroblast growth factors in the domestic dog ovary. Reprod Domest Anim. 2009 Jul;44 Suppl 2(Suppl 2):65-70. doi: 10.1111/j.1439-0531.2009.01424.x. (They used 07 bitches)

Reviewer 4 Report

Comments and Suggestions for Authors

The manuscript submitted for review in this form is not innovative and does not present a new approach to the topic. The manuscript requires significant revision and cannot be published in its current form.

Reviewer's suggestions

Did the Authors assume that uninfected animals were the control group and infected animals were the examined group? If so (since this is not clearly stated in the manuscript), they should refer in the Abstract to the results regarding preantral follicle counts and other parameters analyzed in both study groups. The abstract should reflect the content of the article.

The main problem is the small size of animals used in the study (only 3 uninfected and 2 infected). It is difficult to draw any reliable conclusions from this, especially since there is also a lack of statistical analysis.

On what basis do the Authors believe that these are young individuals? Can their age be determined in any way? I suppose that the Authors determined this based on the morphology of the ovaries of the studied animals and the presence of only preantral follicles. Such information should also be included in the manuscript.

Lines 86–97; In the methodology, the Authors mentioned that they used ELISA to detect bacterial antigens PGL-I and/or LID-1, which showed that 2 of the 5 individuals had infection, but the results of this analysis were not presented. It is incomprehensible that the Authors referred here to a publication (item 17). This publication concerns a much larger group of animals compared to the manuscript submitted for review. The Authors should have used a similar number of armadillos in their studies as well.

Table 3, The Authors presented only the results as means, there is no statistical analysis significantly confirming that infected animals presented larger values ​​for the diameters of the testis, oocyte and follicle than uninfected individuals. A statistical significance of the observed differences is needed

Lines 126-128: The Authors should rather cite the publications of A. Gougeon, G.B.N. Chainy: Morphometric studies of small follicles in the ovaries of women of different ages J. Reprod. Fertil., 81 (1987), pp. 433-442 than Lucci et al. (21)

In addition, the following data are missing from the manuscript: number of sections obtained (from the left/right or both ovaries?) and number of sections observed. The Authors used these data in the formula to calculate the preantral follicle population, but did not include them in the methodology.

Lines 130-131; Are the results given in Table 4 regarding the PE population size (estimated using the given formula, see lines 130-131) expressed as “Total per follicle category”?

If so, this description should be unified and I would suggest using “PE population” as standard notation - according to the data included in the methodology. Such a notation would be clearer to the reader.

Furthermore, what does the parameter Proportions (%) presented in Table 4 refer to? I assume that it is the proportion of a given follicle category to all follicle types in a given ovary. The Authors should explain this clearly.

The presented images (figures 1-4) are of poor quality.

Lines 361-363 (Discussion chapter) - This statement requires confirmation by statistical analysis.

Author Response

Reviewer 4:

Comments 1: The manuscript submitted for review in this form is not innovative and does not present a new approach to the topic. The manuscript requires significant revision and cannot be published in its current form.

Response 1: This is the first study regarding the characterization of the ovarian follicle population in an armadillo species and, therefore, we would like to highlight the novelty of our observations and its contribution to the knowledge of the reproductive physiology of the species.

Comments 2: Did the Authors assume that uninfected animals were the control group and infected animals were the examined group? If so (since this is not clearly stated in the manuscript), they should refer in the Abstract to the results regarding preantral follicle counts and other parameters analyzed in both study groups. The abstract should reflect the content of the article.

Response 2: We have revised the abstract.  Our objective was to provide a descriptive report on the characteristics of the ovarian preantral follicle population in six-banded armadillos. During the epidemiological survey of leprosy carried out in our region, we only had access to the biological material from these 05 individuals, due to logistical issues related to the epidemiological research itself. This limited the possibility of performing statistical analyses of our data to strengthen our conclusions, considering two different groups. At first, we had thought of grouping all the animals into a single group and only providing general data for the species. However, as we had the possible bias of leprosy infection in two animals, we preferred to keep the information separately. We hesitate to distinguish a control vs. an infected group given the small number of individuals. Despite the limitations of the study, which we acknowledge and include in the last paragraph of the discussion, we highlight the important contribution of our study to the knowledge of the reproductive physiology of the species. We also emphasize that when it comes to wild animals, important information about the biology or physiology of the species often comes only from reports, often from a single or few animals, especially for morphological descriptions, as shown in the following articles:

  1. Lopes GP, Santos RR, Almeida DV, Brito AB, Queiroz HL, Domingues SFS. Population estimate and morphometry of ovarian preantral follicles from three recently recognized squirrel monkey species: a comparative study. Zygote. 2017 Jun;25(3):279-287. doi: 10.1017/S0967199417000107. (They used 01 Saimiri microdon, 02 Saimiri cassiquiarensis, and only one ovary from a Saimiri vansolini)
  2. Radcliffe RW, Eyres AI, Patton ML, Czekala NM, Emslie RH. Ultrasonographic characterization of ovarian events and fetal gestational parameters in two southern black rhinoceros (Diceros bicornis minor) and correlation to fecal progesterone. 2001 Mar 15;55(5):1033-49. doi: 10.1016/s0093-691x(01)00464-2. (They used 02 individuals)
  • Klohonatz K, Durrant B, Sirard MA, Ruggeri E. Granulosa cells provide transcriptomic information on ovarian follicle dynamics in southern white rhinoceros. Sci Rep. 2024 Aug 20;14(1):19321. doi: 10.1038/s41598-024-70235-7. (They used 03 individuals)
  1. Rodrigues FR, Da Silva VM, Barcellos JF, Lazzarini SM. Reproductive anatomy of the female Amazonian manatee Trichechus inunguis Natterer, 1883 (Mammalia: Sirenia). Anat Rec (Hoboken). 2008 May;291(5):557-64. doi: 10.1002/ar.20688. (They used 03 individuals)
  2. Kinoshita K, Nakamura T, Kimura K, Shimizu M, Kuze N, Ozaki Y. Gynaecological diagnosis by ultrasound and the measurement of urinary sex steroid hormones in female orangutans. Vet Med Sci. 2020 Aug;6(3):612-616. doi: 10.1002/vms3.237. E (They used 03 individuals)
  3. Scalercio SR, Brito AB, Domingues SF, Santos RR, Amorim CA. Immunolocalization of growth, inhibitory, and proliferative factors involved in initial ovarian folliculogenesis from adult common squirrel monkey (Saimiri collinsi). Reprod Sci. 2015 Jan;22(1):68-74. doi: 10.1177/1933719114532842. (They used 04 individuals)
  • Sattler R, Bishop A, Woodie K, Polasek L. Characterizing estrus by trans-abdominal ultrasounds, fecal estrone-3-glucuronide, and vaginal cytology in the Steller sea lion (Eumetopias jubatus). 2018 Oct 15;120:25-32. doi: 10.1016/j.theriogenology.2018.07.020. (They used 03 individuals)

Comments 3: The main problem is the small size of animals used in the study (only 3 uninfected and 2 infected). It is difficult to draw any reliable conclusions from this, especially since there is also a lack of statistical analysis.

Response 3: As explained in Response 2, our main objective was to provide unreported information on ovarian features from a wild Xenarthra species. We chose not to include a statistical analysis based on a small number of individuals in each group. We made this decision because we believed that statistics based on the number of individuals would not be useful. Descriptive characterizations of the six-banded armadillo ovarian features already provide more reliable information regarding the reproductive physiology of the species, which remains in need of further elucidation. However, we have rewritten some parts of our introduction, as well as added general data for the species to the results tables. In addition, we have modified some aspects of the discussion and conclusion. All modifications are marked in red in the text.

Comments 4: On what basis do the Authors believe that these are young individuals? Can their age be determined in any way? I suppose that the Authors determined this based on the morphology of the ovaries of the studied animals and the presence of only preantral follicles. Such information should also be included in the manuscript.

Response 4: We included the information at the text that “The animals were then characterized as young based on their body weight and based on the histological characteristics of their ovary relative to the emergence of secondary follicles, as previously described for another species of armadillo, Dasypus novemcinctus (Peppler et al., 1985).”

Comments 5: Lines 86–97; In the methodology, the Authors mentioned that they used ELISA to detect bacterial antigens PGL-I and/or LID-1, which showed that 2 of the 5 individuals had infection, but the results of this analysis were not presented. It is incomprehensible that the Authors referred here to a publication (item 17). This publication concerns a much larger group of animals compared to the manuscript submitted for review. The Authors should have used a similar number of armadillos in their studies as well.

Response 5: We would have loved to have used the same number of animals as in the study regarding the epidemiological survey of leprosy in our region. However, the study in question was conducted by the team involved in the diagnosis of infectious diseases, and they had another purpose for using the carcasses of the animals. Therefore, they only provided us with 05 individuals so that we could carry out this report regarding the ovarian features of the species. Furthermore, t is not easy to obtain licenses to capture wild animals in the field for the purpose of studies that involve their euthanasia. Therefore, we took advantage of the opportunity to generate unprecedented data on the reproductive physiology of the species.

Comments 6: Table 3, The Authors presented only the results as means, there is no statistical analysis significantly confirming that infected animals presented larger values for the diameters of the testis, oocyte and follicle than uninfected individuals. A statistical significance of the observed differences is needed.

Response 6: To be consistent with the decision not to perform statistical comparison between groups, we preferred to avoid mentioning that oocyte diameters in infected animals would be higher than those in uninfected animals. Therefore, we tried to exclude such assertions from both the tittle, abstract, results and discussions. We did this to support our intention of demonstrating that our aim was only to characterize, for the first time, the ovarian features of the species.

Comments 7: Lines 126-128: The Authors should rather cite the publications of A. Gougeon, G.B.N. Chainy: Morphometric studies of small follicles in the ovaries of women of different ages J. Reprod. Fertil., 81 (1987), pp. 433-442 than Lucci et al. (21)

Response 7: Thank you for the suggestion.  We substituted the reference as requested.

Comments 8: In addition, the following data are missing from the manuscript: number of sections obtained (from the left/right or both ovaries?) and number of sections observed. The Authors used these data in the formula to calculate the preantral follicle population, but did not include them in the methodology.

Response 8: We added this information to the Material and Methods section as requested.

Comments 9: Lines 130-131; Are the results given in Table 4 regarding the PE population size (estimated using the given formula, see lines 130-131) expressed as “Total per follicle category”?

If so, this description should be unified and I would suggest using “PE population” as standard notation - according to the data included in the methodology. Such a notation would be clearer to the reader.

Response 9: We replaced the description as requested.

Comments 10: Furthermore, what does the parameter Proportions (%) presented in Table 4 refer to? I assume that it is the proportion of a given follicle category to all follicle types in a given ovary. The Authors should explain this clearly.

Response 10: We replaced the description as requested.

Comments 11: The presented images (figures 1-4) are of poor quality.

Response 11: We have included images with better resolution (Figure 2 – 4). Regarding Figure 1, however, we tried to provide a higher resolution image of the animals' reproductive tract, but we were unable to do so. These images were taken on the day the animals were euthanatized, unfortunately, using a low-quality camera. We were not originally going to include this image in the manuscript, but we decided to keep it even though it was not ideal.

Comments 12: Lines 361-363 (Discussion chapter) - This statement requires confirmation by statistical analysis.

Response 12: As previously mentioned, we chose to focus our work on the descriptive characterization of the ovarian features of six-banded armadillos. Thus, we modified the discussion of our work to make it clear that infection by M. leprae did not directly promote major changes in the histological findings in the ovaries of this species.

Reviewer 5 Report

Comments and Suggestions for Authors

A small number of animals - five in total. Subsequently divided into two sroups (uninfected and infected). This minimal number does not allow the authors to be categorical when discussimg the obtained results and to fully fulfill their main aim. When discussing the results, an opinion is most often expressed about the need for future researchq which belittles the research that has been conducted.

Table 2 of the results presented for the total number of multioocyte preantral follicles show a significant differenceq both between infected and uninfected female. But this difference is not commented on.

In conclusion, it is argued that evidence is presented M. leprae infection can clearly affect ovarian health in this s,ecies. The persented results do not allow for such certainty.

Author Response

Reviewer #5

Comments 1: A small number of animals - five in total. Subsequently divided into two groups (uninfected and infected). This minimal number does not allow the authors to be categorical when discussing the obtained results and to fully fulfill their main aim. When discussing the results, an opinion is most often expressed about the need for future research which belittles the research that has been conducted.

Response 1: Initially, our main objective was not to compare the two groups, mainly because we knew that the number of individuals per group would be limited. Our main objective was to characterize the ovarian features in the species for the first time. We only mentioned the two groups because the animals were provided by another epidemiological investigation of leprosy, and they later gave us the results that our sample consisted of 02 infected animals and 03 uninfected animals. Therefore, we thought it was important to highlight this in the work. Based on the group of 05 animals, the methodology used for this purpose of histological characterization and estimation of ovarian follicular population was based on similar previous studies in both wild and domestic animals, which used similar numbers of animals, as mentioned below. In any case, to make our objective of providing general data for the species clearer, we slightly modified the text of the article, with regard to its abstract, objectives, results and discussion.

  1. Lopes GP, Santos RR, Almeida DV, Brito AB, Queiroz HL, Domingues SFS. Population estimate and morphometry of ovarian preantral follicles from three recently recognized squirrel monkey species: a comparative study. Zygote. 2017 Jun;25(3):279-287. doi: 10.1017/S0967199417000107. (They used 01 Saimiri microdon, 02 Saimiri cassiquiarensis, and only one ovary from a Saimiri vansolini)
  2. Rodrigues FR, Da Silva VM, Barcellos JF, Lazzarini SM. Reproductive anatomy of the female Amazonian manatee Trichechus inunguis Natterer, 1883 (Mammalia: Sirenia). Anat Rec (Hoboken). 2008 May;291(5):557-64. doi: 10.1002/ar.20688. (They used 03 individuals)
  • Kinoshita K, Nakamura T, Kimura K, Shimizu M, Kuze N, Ozaki Y. Gynaecological diagnosis by ultrasound and the measurement of urinary sex steroid hormones in female orangutans. Vet Med Sci. 2020 Aug;6(3):612-616. doi: 10.1002/vms3.237. E (They used 03 individuals)
  1. Carrijo OA Jr, Marinho AP, Campos AA, Amorim CA, Báo SN, Lucci CM. Morphometry, estimation and ultrastructure of ovarian preantral follicle population in queens. Cells Tissues Organs. 2010;191(2):152-60. doi: 10.1159/000225935. (They used 05 queens)
  2. Mondadori RG, Santin TR, Fidelis AA, Porfírio EP, Báo SN. Buffalo (Bubalus bubalis) pre-antral follicle population and ultrastructural characterization of antral follicle oocyte. Reprod Domest Anim. 2010 Feb;45(1):33-7. doi: 10.1111/j.1439-0531.2008.01199.x. (They used 05 female buffalos)
  • Warren L, Murawski M, Wilk K, Zieba DA, Bartlewski PM. Suitability of antral follicle counts and computer-assisted analysis of ultrasonographic and magnetic resonance images for estimating follicular reserve in porcine, ovine and bovine ovaries ex situ. Exp Biol Med (Maywood). 2015 May;240(5):576-84. doi: 10.1177/1535370214560971. (They used 05 porcine, 05 ovine and 05 bovine females)
  1. Lucci CM, Amorim CA, Rodrigues AP, Figueiredo JR, Báo SN, Silva JR, Gonçalves PB. Study of preantral follicle population in situ and after mechanical isolation from caprine ovaries at different reproductive stages. Anim Reprod Sci. 1999 Aug 16;56(3-4):223-36. doi: 10.1016/s0378-4320(99)00045-7. (They used 06 does)
  2. Songsasen N, Fickes A, Pukazhenthi BS, Wildt DE. Follicular morphology, oocyte diameter and localisation of fibroblast growth factors in the domestic dog ovary. Reprod Domest Anim. 2009 Jul;44 Suppl 2(Suppl 2):65-70. doi: 10.1111/j.1439-0531.2009.01424.x. (They used 07 bitches)

Comments 2: Table 2 of the results presented for the total number of multioocyte preantral follicles show a significant difference both between infected and uninfected female. But this difference is not commented on.

Response 2: In fact, since the work focuses on a descriptive characterization of the ovarian features of the species, it does not provide statistical analysis of the data, mainly due to the limitation regarding the number of individuals in each group. We believe that the reviewer is referring to the individual variations that were observed for both infected and uninfected animals. Thus, to address this comment, we have included in the appropriate section a brief discussion regarding this individual variation that is also observed in numerous mammalian species.

Comments 3: In conclusion, it is argued that evidence is presented M. leprae infection can clearly affect ovarian health in this species. The presented results do not allow for such certainty.

Response 3: We agree with the reviewer and therefore try to modify this consideration throughout the manuscript, including in the conclusion.

Round 2

Reviewer 3 Report

Comments and Suggestions for Authors

I still think that with one or two individuals in each subgroup, the data are highly contingent

Reviewer 4 Report

Comments and Suggestions for Authors

Despite the proofreading of the manuscript by the Authors, the Reviewer did not receive satisfactory answers to the previous comments. Therefore, the Reviewer maintains the decision to reject the manuscript.

Comments:

Lines 58-60. It is said: The main aim of the study was to characterize and estimate the ovarian preantral  follicle population in juvenile six-banded armadillos. In addition, we had the opportunity to describe the ovarian features in individuals infected or not infected with M. leprae.

 In my opinion it would be enough to write:  In addition, we had the opportunity to describe the ovarian features in individuals infected with M. leprae.

Linies 82-93: Detection of M. Leprae infection. The Reviewer did not receive a satisfactory response from the Authors. I do not understand the point of including a description of this part of the experiment if the data from these analyses are not included in the manuscript. The authors explained that they obtained such information from another research group. Nevertheless, such data should definitely be included in this manuscript (for example, in the form of a table). I believe that such data certainly exists.

Line 113. It is said: For morphometry analysis, approximately 30 follicles were evaluated for each  category (primordial, primary and secondary) by light microscopy.

However, the number of secondary follicles analyzed is variable (from 10 to 28), see Table 3. To avoid such ambiguities, it should be noted in the Materials and Methods section that the number of follicles analyzed is given in Table 3.

In my opinion, the data in Table 3 can be analyzed statistically. In this case, the group size (about 30 follicles of each type) is sufficient for statistical analysis.

See the following article:

Lopes GP, Santos RR, Almeida DV, Brito AB, Queiroz HL, Domingues SFS. Population estimate and morphometry of ovarian preantral follicles from three recently recognized squirrel monkey species: a comparative study. Zygote. 2017 Jun;25(3):279-287. doi: 10.1017/S0967199417000107.

The Authors concluded that there are no significant differences in the ovarian parameters studied between infected and uninfected females. However, Table 5 clearly shows the differences in the number of healthy and degenerated follicles in M. leprae-infected and uninfected females. Based on the data presented in Table 5, it can be concluded (or assumed) that there is increased follicular atresia in infected individuals. However, to confirm this, studies on a larger number of animals (at least 4) and, of course, statistical analysis are necessary.

 In my opinion, it is incorrect to assume that infected and uninfected individuals are treated as one group for which an overall mean (for normal and degenerated proportion of follicles %) is calculated.